# Transient Temperature Distributions on Lithium-Ion Polymer SLI Battery

**Yiqun Liu, Y. Gene Liao * and Ming-Chia Lai**

Collage of Engineering, Wayne State University, Detroit, MI 48202, USA
* Correspondence: geneliao@wayne.edu; Tel.: +1-313-577-8078

**Abstract:** Lithium-ion polymer batteries currently are the most popular vehicle onboard electric energy storage systems ranging from the 12 V/24 V starting, lighting, and ignition (SLI) battery to the high-voltage traction battery pack in hybrid and electric vehicles. The operating temperature has a significant impact on the performance, safety, and cycle lifetime of lithium-ion batteries. It is essential to quantify the heat generation and temperature distribution of a battery cell, module, and pack during different operating conditions. In this paper, the transient temperature distributions across a battery module consisting of four series-connected lithium-ion polymer battery cells are measured under various charging and discharging currents. A battery thermal model, correlated with the experimental data, is built in the module-level in the ANSYS/Fluent platform. This validated module thermal model is then extended to a pack thermal model which contains four parallel-connected modules. The temperature distributions on the battery pack model are simulated under 40 A, 60 A, and 80 A constant discharge currents. An air-cool thermal management system is integrated with the battery pack model to ensure the operating temperature and temperature gradient within the optimal range. This paper could provide thermal management design guideline for the lithium-ion polymer battery pack.

**Keywords:** battery heat; battery temperature; battery thermal management; battery thermal model; lithium-ion battery

---

## 1. Introduction

Lithium-ion batteries have become the most popular electric energy storage systems for many configurations of hybrid and electric vehicles due to their high energy-to-weight ratio, high energy-to-volume ratio, and excellent cycle life [1]. The lithium-ion battery is also a good candidate to replace the traditional lead-acid starting, lighting, and ignition (SLI) battery [2]. One of the major concerns in using lithium-ion batteries is that operating temperature has a significant impact on performance, safety, and cycle lifetime of the batteries. The typically acceptable operating temperature region for lithium-ion batteries is between −20 °C and 60 °C, while an optimal temperature range is from 15 °C to 35 °C. Furthermore, the maximum temperature difference from cell to cell in a module should not exceed 5 °C to avoid a severe temperature gradient. The battery usable capacity and discharging voltage are decreased under low operating temperature. The cycle life of the battery is shortened under high operating temperature. Some safety issues might be initiated under high operating temperature. To maintain the operating temperature in the optimal range, a thermal management system (TMS) is used to cool down or warm up the battery. Understanding the heat generation characteristics and temperature distributions of the lithium-ion batteries is important to design an effective TMS.

Heat generated in the lithium-ion battery is mainly from two processes: entropic heat generation due to changes in electrochemical reactions, and ohmic heating resulting from current flow through cell internal resistance during charge and discharge process. Equation (1) expresses the heat generation

rate per unit volume $\dot{q}$ ($W/m^3$) of the lithium-ion battery cell [3]. The first term and the second term on the right-side of Equation (1) are the ohmic heating and entropic heat generation, respectively.

$$\dot{q} = i^2 R + iT \frac{dE_{OC}}{dT} \tag{1}$$

where $i$ ($A/m^3$) is the volumetric current density (positive value for discharge and negative value for charge), $R$ ($\Omega$) is internal resistance of battery cell, $T$ (K) is the temperature, and $dE_{OC}/dT$ ($V/K$) is the temperature coefficient at the open circuit voltage (OCV) of the battery cell. Research shows that up to 54% of the total heat generated in the lithium-ion battery cell is contributed from the ohmic heating [4]. The lithium-ion battery cells generate more heat during the high current charge and discharge process. Accumulated heat in certain areas of the battery pack will lead to temperature nonuniformity among the cells. The non-uniform temperature across the battery pack results in different performance, self-discharging rate, state-of-health, and state-of-life of each individual cell in the battery pack. Research shows that 10 °C to 15 °C temperature gradient across the battery pack causes a 30% to 50% degradation of the cells [5,6]. Therefore, a well-designed thermal management system is important to maintain each cell temperature within the optimal range and keep the temperature uniform across the battery pack [7]. There are three types of thermal management systems for battery pack: air cooling [8,9], liquid cooling [10,11], and phase change material (PCM) cooling [12,13]. The air cooling and liquid cooling are commonly used for traction battery packs in hybrid and electric vehicles. The air cooling has advantages of cost and space while the cooling efficiency is not as good as liquid cooling. The liquid cooling usually costs more and occupies more space due to the requirement of liquid tank, hydraulic pump, complex piping layout, and heat exchanger.

This paper investigates transient temperature distributions on a lithium-ion polymer SLI battery. The transient temperatures of a SLI battery module, consisting of four series-connected lithium-ion polymer cells, are measured during different charge and discharge experiments. A numerical thermal model of this battery module is built in the ANSYS/Fluent platform and the simulated temperature data is correlated with the experimental data. Four of these battery modules are parallel-connected to form a SLT battery pack model with 14.4 V nominal voltage and 20 Ah capacity. Heat generation characteristics and transient temperatures in the pack are simulated. Finally, a thermal management system using air cooling is simulated for this battery pack. This paper could provide thermal management design guideline for the lithium-ion polymer battery pack.

## 2. Temperature Experiments of Battery Module

The proposed lithium-ion polymer SLI battery are constructed by four parallel-connected modules. Each module consists of four series-connected EiG-ePLB-C020 lithium-ion polymer cells which are labeled as Cell-A, B, C, and D as shown in Figure 1. Each cell, $LiNiCoMnO_2$-based cathode and graphite-based anode, has 3.6 V nominal voltage and 20 Ah capacity such that the battery module is capable of 14.4 V nominal voltage and 20 Ah capacity. Each cell is fixtured within a protection case. One side of the protection case is steel plated and the other side is open to the space. Each protection case is mounted to the bottom base plate by two screws. For easier series-connection, neighbor cells are mounted in the opposite orientation where the positive terminal of one cell is close to the negative terminal of its neighbor cell. Six small copper plates are used to connect the four cells on the top of protection cases. The gaps between two neighbor cells are 7 mm. The cell surface temperatures are measured by eight OMEGA surface thermocouples. Thermocouple A1, A2, A3, and A4 are adhesive on one side of the Cell-A which is facing to open space. Thermocouple B1, B2, B3, and B4 are on one side of the Cell B which is facing Cell-C. These thermocouples are connected to two TEKCOPLUS 4-channel K-type thermocouple meters.

The experiments are conducted in the Envirotronics temperature chamber and the temperature is set to 25 °C (298 K). the charge and discharge tests are conducted using the Digatron Charge/Discharge Unit which is controlled by the Battery Manager 4 software. Temperatures at eight thermocouples are

obtained during 20 A, 40 A, 60 A, and 80 A continuous constant current discharge tests as well as 20 A and 40 A continuous constant current charge tests. To prevent the cells from being overcharged or over-discharged, the maximum charging voltage and minimum discharging voltage of the battery module are set to 16.4 V and 10 V, respectively. Each cell in the battery module is fully charged to 4.17 V and completely cooled down to 25 °C before starting the discharge test. Similarly, each cell is fully discharged to 2.5 V and completely cooled down to 25 °C before starting the charge test.

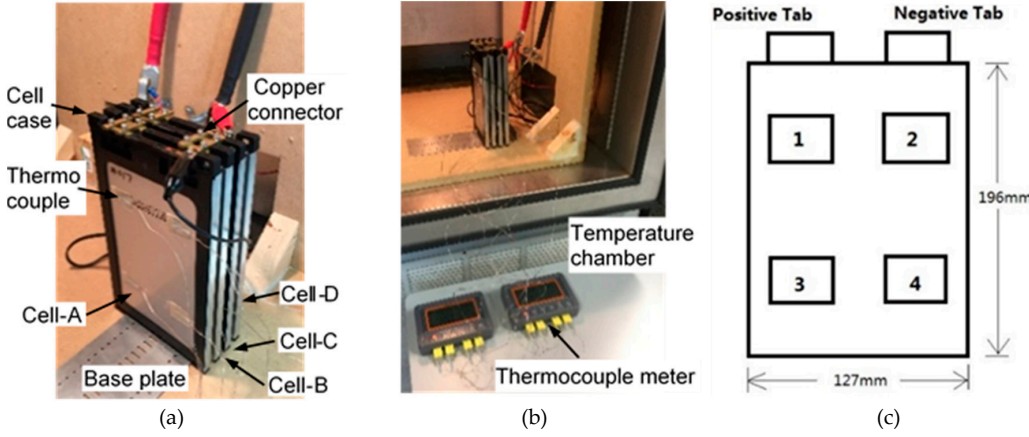

**Figure 1.** Experimental setup: (**a**) battery module layout; (**b**) temperature chamber setup; (**c**) thermocouple setup.

Figure 2 shows the temperatures measured by thermocouples placed on eight locations within the battery module during different charge and discharge tests. During the 20 A constant current charge test, temperatures at all eight locations are very close to each other and the maximum temperature difference is 0.5 K. All temperatures increase during the test, but the increments are only about 0.7 K. During the 20 A discharge test, all temperatures are also very close to each other and the maximum difference is about 0.5 K. The temperature difference becomes larger at the end of the test and is about 1 K. During the first 40 min, the temperature at each location only changes very slightly. All temperatures start to increase after 40 min and increase about 1 K before the end of the test.

During the 40 A charge test, temperatures at all locations increase quickly at the beginning of the test. The increasing rates become slower as the tests go on. After the first 10 min of test, all temperatures at the locations on the Cell-B are higher than those at the corresponding locations on the Cell-A. At the end of the test, the average temperature of locations B1 to B4 is 1 K higher than that of locations A1 to A4. Cell-B and C are in the middle of module assembly, so the heat generated at locations B1 to B4 is difficult to dissipate. Locations A1 to A4 are facing to open space with a constant temperature of 298 K which is controlled by the temperature chamber. Therefore, the heat generated at locations A1 to A4 easily dissipate. During the 40 A discharge test, temperatures measured on the Cell-B is higher than that measured on Cell-A in the first 7 min of the test. After the first 7 min, the temperatures at the locations B1 to B4 increase much quickly than the temperatures at the locations A1 to A4. Approaching to the end of test, temperatures at all eight locations start to increase rapidly. The largest difference of average temperature between B1 to B4 and A1 to A4 is about 2 K. During the 60 A and 80 A discharge tests, temperatures measured on the Cell-B become higher than those on Cell-A after 4 min. The largest difference of average temperature between locations B and A is about 5 K for both 60 A and 80 A discharge tests.

The experimental results indicate that temperature increments (from start to end of tests) at each location become larger as the charge/discharge current increases. The increment on average temperature of eight locations is 0.6 K, 1.6 K, 5.3 K, 9.8 K, 15.8 K, and 20.7 K during the 20 A charge, 20 A discharge, 40 A charge, 40 A discharge, 60 A discharge, and 80 A discharge test, respectively. The temperature difference between Cell-A and Cell-B becomes larger with the increase of the charging/discharging current.

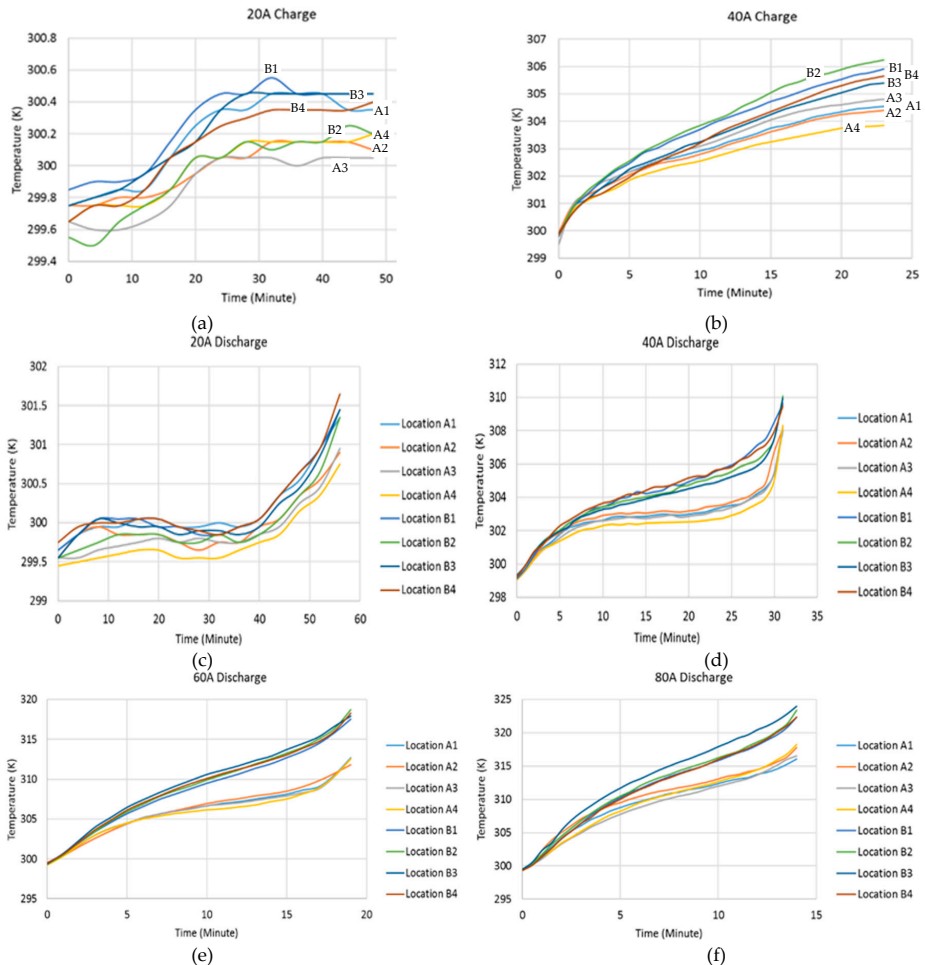

**Figure 2.** Temperatures measured by eight thermocouples within the battery module: (**a**) 20 A charge; (**b**) 40 A charge; (**c**) 20 A discharge; (**d**) 40 A discharge; (**e**) 60 A discharge; (**f**) 80 A discharge.

## 3. Thermal Model of Battery Module

The battery module thermal model is built in the ANSYS/Fluent platform. A semi-empirical electrochemical model based on the dual-potential approach of Newman, Tiedemann, Gu, and Kim (NTGK) is selected. Kwon [14] proposed the NTGK model and later researchers [15–17] used and validated the NTGK model. The current transfer $j$ and the potential field $(\varphi_+ - \varphi_-)$ is related by the Equation (2) in the model.

$$j = aY[U - (\varphi_+ - \varphi_-)] \tag{2}$$

where $a$ is the specific area of the electrode sandwich sheet, $\varphi_+$ and $\varphi_-$ are respectively phase potentials of the positive and negative electrodes. $Y$ and $U$ are the NTGK model parameters and can be determined by the curve fitting of the voltage-current response curve from the testing for a certain battery. Equations (3) and (4) [14] are used to calculate the values of $Y$ and $U$.

$$Y = \left(\sum_{n=0}^{5} a_n (DOD)^n\right) \exp\left[-C_1\left(\frac{1}{T} - \frac{1}{T_{ref}}\right)\right] \tag{3}$$

$$U = \left(\sum_{n=0}^{3} b_n (DOD)^n\right) - C_2\left(T_{rt} - T_{ref}\right) \tag{4}$$

where $C_1$ and $C_2$ are the NTGK model constants, $a_n$ and $b_n$ are the coefficients used to calculate $Y$ and $U$ in [18], $T_{rt}$ is the real-time temperature, $T_{ref}$ is the reference temperature (298 K), DOD is the depth of discharge and can be calculated by using Equation (5).

$$DOD = \frac{Vol}{3600Q_{Ah}}\left(\int_0^t j\,dt\right) \tag{5}$$

where *Vol* is the volume of the battery, and $Q_{Ah}$ is the total capacity (Ah) of the battery. Equation (6) shows the energy sources including the Joule heating, electrochemical reaction heating, and entropic heating:

$$\dot{q} = \sigma_+\nabla^2\varphi_+ + \sigma_-\nabla^2\varphi_- + j\left[U - (\sigma_+ - \sigma_-) - T\frac{dU}{dT}\right] \tag{6}$$

where $\sigma_+$ and $\sigma_-$ are respectively the effective electric conductivities for the positive and negative electrodes.

The procedures of the ANSYS/Fluent modeling and simulation for the battery module temperature distributions are listed below [19,20].

(1) In the DesignModeler, the 3D model of the battery module is created according to the actual size of the EiG ePLB-C020 cell and the actual configuration of the battery module. The dimension sizes of the cell are 127 mm (width) × 196 mm (length) × 7 mm (thickness). The dimensions for the positive and negative tabs are 30 mm (width) × 23 mm (length) × 6 mm (thickness). The dimensions for the three connecting plates are 30 mm (width) × 21 mm (length) × 5 mm (thickness). Also, the gaps between two neighbor cells are 7 mm. The protection cases of the cells are not modeled in the simulation to reduce the complexity of the simulations.

(2) An air volume of 127 mm × 196 mm × 7 mm is created in the space between two neighboring cells, so the space between neighboring cells is completely filled with air.

(3) The mesh of the entire geometry is created with 165,049 nodes and 132,761 elements.

(4) The mesh file is input to the Fluent Setup where the Energy and the Multi-Scale Multi-Domain (MSMD) Battery Model are activated. Considering the tradeoff between accuracy and complexity of all models, the simple semi-empirical electrochemical NTGK Empirical Model is selected in the MSMD Battery Model dialog box. A value of 20 Ah is entered for the Nominal Cell Capacity and specified C-rate is selected for the Solution Option. Positive (for discharge) and negative (for charge) C-rate values can be entered for different tests. The values of 10 V and 16.4 V are entered for the Min. and Max. Stop Voltages, respectively.

(5) Under model parameters, initial DOD is set to 0 for discharging simulations and Initial DOD is set to 1 for charging simulations. All the Y and U Coefficients are determined by the method provided in [15]. Under conductive zones, active components, tab components, and busbar components are assigned. Under electric contacts, the negative tab and positive tab for the entire battery module are assigned.

(6) Copper is selected as the tab and connecting plate material in the Materials input. For the active material inside the battery cell, 2092 kg/m$^3$, 678 J/kg·K, 18.4 W/m·K, and 3.541 × 10$^7$ siemens/m are entered for the density, specific heat, thermal conductivity, and electrical conductivity, respectively. The uds-0 and uds-1 coefficients are respectively set to 1190,000 kg/m·s and 983,000 kg/m·s. Air is the selected material for space between two neighbor cells.

(7) The thermal properties of battery cell case walls, positive/negative tabs, and connecting plates are edited in the boundary conditions. The thermal condition is set as mixed for all zones. The heat transfer coefficient is 5 W/m$^2$-K and external emissivity is 0.9. The free stream temperature is set to 298 K which is the temperature inside the temperature chamber.

(8) The SIMPLE scheme is selected as solution methods. The hybrid is the initialization method. An initial temperature is set to 298 K. A fixed time stepping method is used under run calculation. Time step size is set to 60 s and the number of time steps are determined based on the different C-rates.

Figure 3 shows the simulated temperature distributions at the end of the 80 A discharge on the module which consists of four series-connected cells. From left to right, these four cells are labeled as Cell-A, B, C, and D as shown in Figure 1a. Cell-B and C have higher surface temperatures than Cell-A

and D do. The module has symmetric assembly such that Cell-A and D have the same temperature distributions as well as Cell-B and C have the same temperature distributions. It is sufficient to investigate the simulated surface temperature distributions only on Cell-A and B at the end of each simulation, as shown in the Figure 4.

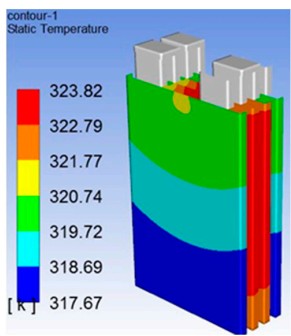

**Figure 3.** Simulated temperature distributions on the module at the end of the 80 A discharge.

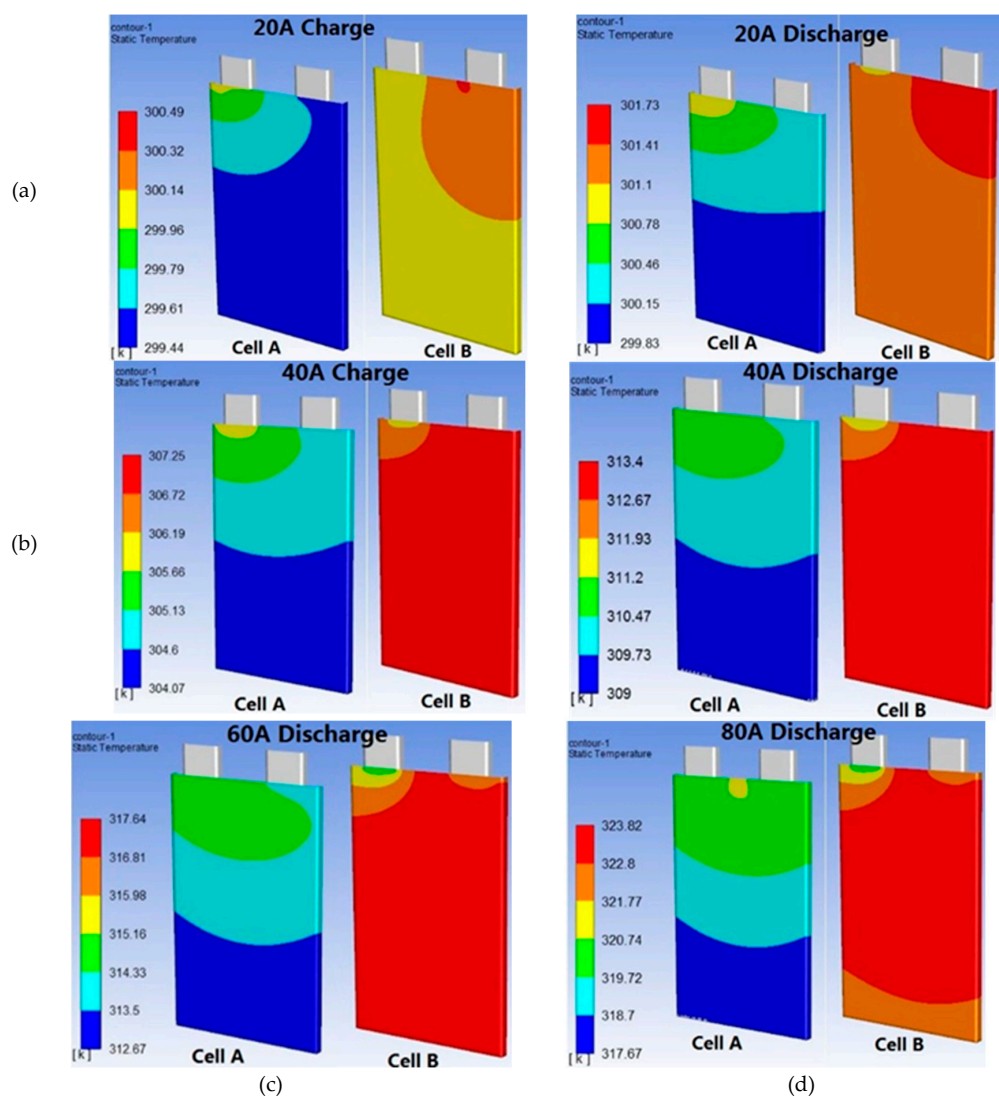

**Figure 4.** Simulated temperature distributions on Cell-A and B at the end of each charge and discharge: (**a**) 20 A; (**b**) 40 A; (**c**) 60 A discharge; (**d**) 80 A discharge.

## 4. Thermal Model Correlations

The simulated cell surface temperatures are correlated with the experimental temperatures measured by thermocouples. The simulation results present that cell surface temperature increases when the discharge current increases. As an example shown in Figure 4, the highest temperature on cell-B increases from 301.73 K to 323.82 K when the discharge current changes from 20 A to 80 A. The largest temperature gradient among the battery module also increases from 1.9 K to 6.15 K as the discharge current increases from 20 A to 80 A. On Cell-A, the temperatures in the neighborhood of tabs always are higher than temperatures on areas away from the tabs. The temperatures are gradually decreased away from the tabs. The Cell-B has the similar phenomenon during the 20 A charge and discharge current. While using higher discharge currents on the Cell-B, areas close to the tabs have lower temperatures than other areas. This inconsistency might be attributed to much heat generated with higher discharging currents and heat accumulated within the middle cells like Cell-B. Consequently, the tabs might function as cooling fins dissipating heat from the top areas. At the end of the 80 A discharge simulation for the Cell-B, the temperatures on the bottom areas are slightly lower than those on the middle areas. This situation could be understood that base plate might also act as cooling fins to dissipate heat from the bottom areas.

Table 1 shows the comparisons between simulated (SIM) and experimental (EXP) surface temperatures on eight locations. The data in the DIF column is the difference between simulation and experiment, or the value of SIM subtracting EXP. Data in the MID row is the temperature data at the midpoint of the charge or discharge process and data in the END row is the temperature data at the end of the process. The average value of the absolute DIF becomes larger with increasing discharge currents. These average values are, respectively, 1.9 K and 1.5 K for 40 A and 80 A discharge. The SLT battery module thermal model is validated with acceptable discrepancy.

**Table 1.** Comparisons between simulated and experimental surface temperatures on eight locations.

| Location | Time | 20 A CHARGE | | | 40 A CHARGE | | | 20 A DISCHARGE | | | 40 A DISCHARGE | | | 60 A DISCHARGE | | | 80 A DISCHARGE | | |
|---|---|---|---|---|---|---|---|---|---|---|---|---|---|---|---|---|---|---|---|
| | | SIM | EXP | DIF | SIM | EXP | DIF | SIM | EXP | DIF | SIM | EXP | DIF | SIM | EXP | DIF | SIM | EXP | DIF |
| Location A1 | MID | 299.5 | 300.4 | −0.9 | 303.8 | 303.2 | 0.6 | 300.1 | 300.0 | 0.1 | 304.3 | 302.9 | 1.4 | 306.2 | 306.4 | −0.2 | 313.9 | 310.8 | 3.1 |
| | END | 299.7 | 300.4 | −0.7 | 305.1 | 304.5 | 0.6 | 300.4 | 301.4 | −1.0 | 310.3 | 308.0 | 2.3 | 314.3 | 312.6 | 1.7 | 320.1 | 315.8 | 4.3 |
| Location A2 | MID | 299.5 | 300.1 | −0.6 | 303.9 | 303.0 | 0.9 | 300.1 | 299.7 | 0.4 | 304.1 | 303.1 | 1.0 | 306.3 | 306.5 | −0.2 | 313.3 | 311.4 | 1.9 |
| | END | 299.6 | 300.1 | −0.5 | 304.8 | 304.4 | 0.4 | 300.1 | 300.9 | −0.8 | 310.0 | 308.1 | 1.9 | 314.2 | 311.8 | 2.4 | 320.0 | 317.9 | 2.1 |
| Location A3 | MID | 299.5 | 300.1 | −0.6 | 303.5 | 303.4 | 0.1 | 300.0 | 399.8 | 0.2 | 303.8 | 302.8 | 1.0 | 305.5 | 306.4 | −0.9 | 312.1 | 310.0 | 2.1 |
| | END | 299.5 | 300.1 | −0.6 | 304.4 | 304.8 | −0.4 | 300.0 | 301.0 | −1.0 | 309.4 | 308.0 | 1.4 | 313.2 | 312.5 | 0.7 | 318.7 | 316.2 | 2.5 |
| Location A4 | MID | 299.5 | 300.1 | −0.6 | 302.5 | 302.8 | −0.3 | 399.8 | 299.5 | 0.3 | 303.8 | 302.6 | 1.2 | 305.6 | 306.1 | −0.5 | 312.2 | 310.3 | 1.9 |
| | END | 299.5 | 300.2 | −0.7 | 304.3 | 303.9 | 0.4 | 399.9 | 300.8 | −0.9 | 309.4 | 308.3 | 1.1 | 313.2 | 312.6 | 0.6 | 318.8 | 318.2 | 0.6 |
| Location B1 | MID | 299.9 | 300.5 | −0.6 | 305.0 | 304.1 | 0.9 | 300.2 | 299.8 | 0.4 | 305.8 | 304.4 | 1.4 | 309.1 | 308.6 | 0.5 | 314.0 | 313.0 | 1.0 |
| | END | 300.0 | 300.4 | −0.5 | 306.9 | 306.0 | 0.9 | 301.2 | 301.5 | −0.3 | 313.2 | 309.7 | 3.5 | 317.5 | 317.3 | 0.2 | 323.2 | 322.3 | 0.9 |
| Location B2 | MID | 300.0 | 300.1 | −0.1 | 304.7 | 304.3 | 0.4 | 300.4 | 299.8 | 0.6 | 306.0 | 304.2 | 1.8 | 309.2 | 309.3 | −0.1 | 314.2 | 313.4 | 0.8 |
| | END | 300.2 | 300.2 | 0.0 | 307.0 | 306.2 | 0.8 | 301.5 | 301.4 | 0.1 | 313.2 | 310.1 | 3.1 | 317.6 | 318.5 | −0.9 | 323.4 | 323.6 | −0.2 |
| Location B3 | MID | 299.9 | 300.4 | −0.5 | 304.2 | 303.7 | 0.5 | 300.2 | 299.8 | 0.4 | 305.7 | 303.9 | 1.8 | 308.7 | 310.0 | −1.3 | 313.8 | 314.6 | −0.8 |
| | END | 300.0 | 300.5 | −0.5 | 306.8 | 305.4 | 1.4 | 301.2 | 301.5 | −0.3 | 312.9 | 310.1 | 2.8 | 317.1 | 317.6 | −0.5 | 322.9 | 324.0 | −1.1 |
| Location B4 | MID | 299.9 | 300.3 | −0.4 | 304.2 | 303.8 | 0.4 | 300.0 | 299.8 | 0.2 | 305.6 | 304.8 | 0.8 | 308.9 | 309.8 | −0.9 | 313.6 | 312.9 | 0.7 |
| | END | 300.1 | 300.4 | −0.3 | 306.9 | 305.6 | 1.3 | 301.3 | 301.7 | −0.4 | 312.8 | 309.5 | 3.3 | 317.2 | 318.2 | −1.0 | 323.0 | 322.3 | 0.7 |
| Average | | | 0.6 | | | 0.6 | | | 0.5 | | | 1.9 | | | 0.8 | | | 1.5 | |

## 5. Extension to a Starting, Lighting, and Ignition (SLI) Pack Thermal Model

Nowadays, a typical SLI battery capacity has increased to 60 Ah. Some vehicles equipped with the remote-control systems (engine starting, seat heating, air conditioning, vehicle status monitoring, etc.) require larger capacity batteries to support these accessories since these features draw electricity from the battery even when the vehicle is parked [2]. An 80 Ah, 14.4 V SLI battery pack formed by four parallel-connected modules is modelled in the ANSYS/Fluent platform. The temperature distributions on the battery pack are investigated under several charge and discharge currents.

Figure 5 shows the temperature distributions on the battery pack at the end of 40 A, 60 A, and 80 A discharge simulations. The areas close to the battery tabs have lower temperature distributions because of the cooling fin effects of the tabs and connecting plates. Two cells in the middle of the battery pack have the highest temperature distributions because the accumulated heat in the middle of the pack is very difficult to be dissipated. Temperatures on the cells located on both open ends are lower than those on the middle cells with about 2 K for 40 A/60 A and 4 K for 80 A discharges. The largest temperature gradient across the pack is about 10 K for all 40 A, 60 A, and 80 A discharges. A large temperature gradient across the battery pack might lead to different degradation rates and unbalancing among cells. An effective thermal management system might be needed for the lithium-ion polymer SLI battery to keep the operating temperature in an optimum range and minimize the temperature gradient across the pack.

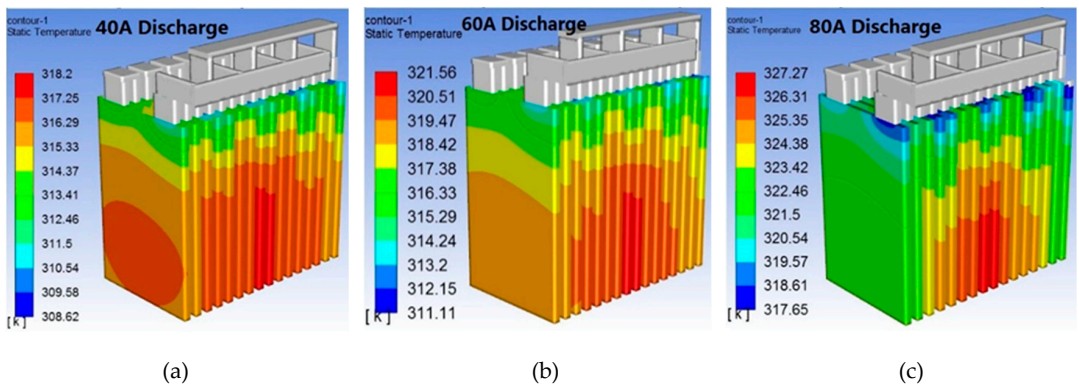

(a)                                                    (b)                                                    (c)

**Figure 5.** Temperature distributions on battery pack at the end of discharges: (**a**) 40 A; (**b**) 60 A; (**c**) 80 A.

## 6. Integration Cooling System with the SLI Pack Thermal Model

This study uses ANSYS/Fluent computational fluid dynamics capability to integrate the developed battery pack thermal model with thermal management system. Due to spaces between adjacent cells in the pack, each cell has two air flows on each side of the cell with one virtual cooling air channel. Two cooling air channels are modeled for each cell, meaning each cell is surrounded by the air flows. Two cooling channels and the cell forms a sandwich-type structure, as shown in Figure 6. The cool air flows enter to the bottom of the pack and exit from the tabs. The dimension sizes of each air inlet and outlet channel is 3.5 mm (or a half-size of the space gap between cells) by 127 mm and 196 mm long, whose area is enough to cover one side area of a cell. The velocity inlets and pressure outlets are set for the cooling channels. The inlet and outlet temperature are, respectively, specified at 288 K (15 °C) and 298 K (25 °C). The air flow velocity can be adjusted based on the different temperature distribution in each charge or discharge simulation, such as the air flow velocity is set to a higher value for the larger current discharges. The thermal condition for the battery cell walls is set to convection with heat transfer coefficient 5 W/m$^2$-K.

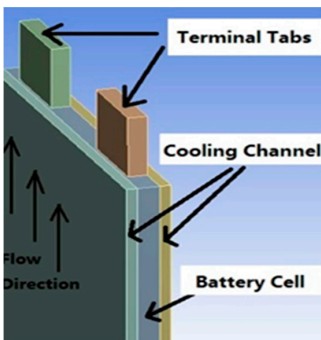

**Figure 6.** Cell with cooling channels in the battery pack.

Figure 7 shows the temperature distribution on the battery pack with cooling system at the end of 40 A, 60 A, and 80 A discharge simulations. The cooling air flow velocity in the channel is set to constant 2m/s for all 40 A, 60 A, and 80 A discharge simulations. The constant air flow velocity removes accumulated heat in the middle of the battery pack such that temperature variation among cells is much smaller than without cooling effect. The temperature variations among cells are, respectively, about 0.5 K, 0.5 K, and 0.7 K for 40 A, 60 A, and 80 A discharges. Also, the temperature gradient within each single cell is smaller comparing with to the cell without cooling system. The largest temperature gradient on a single cell in the battery pack is, respectively, about 3.1 K, 3.7 K, and 4.5 K for the 40 A, 60 A, and 80 A discharges. With the help of the cooling system, the temperature distributions could be uniformly across the entire battery pack. The cooling system also maintains the battery pack operating temperature within a safe range. When discharge current is higher than 80 A, the cooling air velocity should be increased to remove much heat from the battery pack.

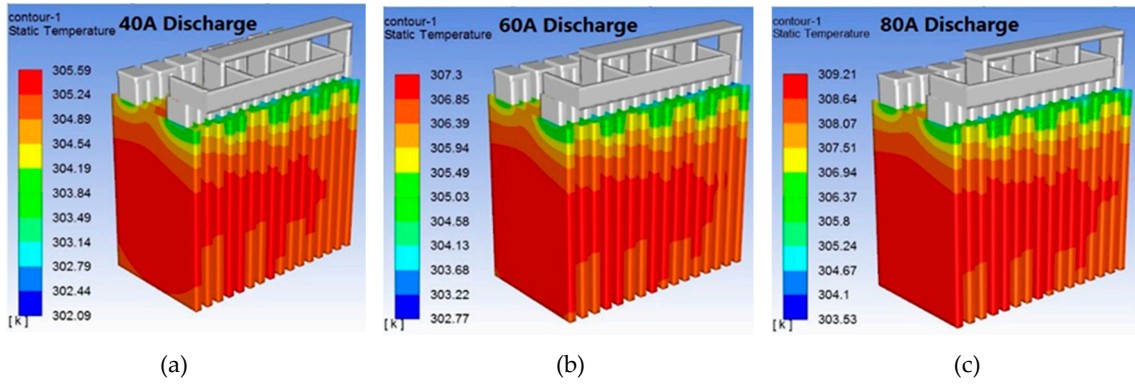

(a)        (b)        (c)

**Figure 7.** Temperature distribution on battery pack with cooling system at the end of discharges: (**a**) 40 A; (**b**) 60 A; (**c**) 80 A.

## 7. Conclusions

Lithium-ion polymer batteries currently are the most popular vehicle onboard electric energy storage systems ranging from the 12 V/24 V SLI battery to the high-voltage traction battery pack in hybrid and electric vehicles. The operating temperature has a significant impact on the performance, safety, and cycle lifetime of lithium-ion batteries. It is essential to quantify the heat generation and temperature distribution of the battery cell, module, and pack to design an effective thermal management system. This paper experimentally and numerically investigates the transient temperature distributions on the SLI battery module and pack. The cell surface temperatures within a module are measured by eight surface thermocouples in a temperature chamber. A battery thermal model, correlated with the experimental data, in the module level is built in the ANSYS/Fluent platform. This validated module thermal model is then extended to a SLI pack thermal model. The temperature

distributions on the battery pack model is simulated under 40 A, 60 A, and 80 A constant discharge currents. An air-cool thermal management system is integrated with the battery pack model to ensure the operating temperature and temperature gradient are within the optimal range.

**Author Contributions:** Experiment, Software Simulation, Y.L.; Paper Writing, Y.L., Y.G.L.; Paper Review and Editing, Y.G.L., M.-C.L.

**Funding:** This work was supported in part by the National Science Foundation, ATE: Centers, under grant number DUE-1801150.

**Conflicts of Interest:** The authors declare no conflict of interest.

## Nomenclature

| | |
|---|---|
| $i$ | volumetric current density (A/m$^3$) |
| $R$ | internal resistance of battery cell ($\Omega$) |
| $T$ | temperature (K) |
| $dE_{OC}/dT$ | temperature coefficient ($V/K$) |
| OCV | open circuit voltage |
| $a$ | specific area of the electrode sandwich sheet |
| $a_n, b_n$ | coefficients used to calculate $Y$ and $U$ |
| $C_1, C_2$ | NTGK model constants |
| DOD | depth of discharge |
| $Q_{Ah}$ | battery total capacity (Ah) |
| $T_{ref}$ | reference temperature (298 K) |
| $T_{rt}$ | real-time temperature |
| $Vol$ | volume of the battery |
| $Y, U$ | NTGK model parameters |
| $\varphi_+, \varphi_-$ | phase potentials of the positive and negative electrodes |
| $\sigma_+, \sigma_-$ | effective electric conductivities for the positive and negative electrodes |

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
