# Peer review of "Transient Temperature Distributions on Lithium-Ion Polymer SLI Battery"

_vehicles, doi:10.3390/vehicles1010008_

Round 1
Reviewer 1 Report
Thank you for your exciting research performance. I shall recommend few comments to improve your paper.your
1. I can not catch your research originality. As you know, there are many previous research works on thermal modeling of lithium-ion battery. You should present your main key research points.
2. Figure 2 shows temperature profiles of batteries. Color plots are very difficult distinguish from each other. Please find any idea to compare each plots.
3. You should define all symbols of equation and figures in Nomenclature.
4. In section 4, you defined title as "Thermal Model Correlations". But I could not catch what kind correlations mean.
5. You tried to explain your thermal models. I expected that your models are "conduction" and "convection" I recommend to explain or define your models clearly.
6. I can not catch your model as a new battery thermal model. You used ANSYS battery module and You used conventional battery pack to simulate. What is your originality to verify your work as a research achievement.
Author Response
The reviewer’s comments and suggestions are highly appreciated, and suggested modifications and improvements have been made in the revised paper. Point-to-point responses and discussions to the reviewer’s questions/suggestions are presented as follows:
1. Thanks for your insightful comments. The originality of this manuscript is experimental and numerical investigation of transient temperature distributions on a large-size lithium-ion polymer battery cell and module. The validated heat generation characteristics and temperature distributions are key data in design of battery thermal management system.
2. The location labels were added to curves in Fig. 2(a) and (b).
3. Thanks for your insightful comments. Nomenclatures were added in lines 362 to 378.
4. A sentence was added in the beginning of the paragraph as “The simulated cell surface temperatures are correlated with the experimental temperatures measured by thermocouples.”
5. Thanks for your insightful comments. Few sentences were added to the battery model setup procedures. The thermal condition used in the cell model is a compound process of convection and radiation. The “conduction” is a built-in function in the ANSYS Fluent. Heat can transfer through any contacting surfaces and users do not need to add “conduction” into the model. In battery model setup procedures, procedure (2) was added in lines 181 to 182 as:
“(2) An air volume of 127mm x 196mm x 7mm is created in the space between two neighbor cells, so the space between neighbor cells is completely filled with air.”
One sentence was added in procedure (6), line 200 as:
“Air is the selected material for space between two neighbor cells.”
Reviewer 2 Report
In this paper, the heat production and temperature rise trajectory of lithium-ion batteries under fixed current are described. The results obtained are credible, and the results obtained in the analysis of transient temperature rise and uniformity of distribution of batteries are prominent.
In the calculation of temperature rise of natural heat dissipation, it would be better if it could be connected with the actual working condition of the battery of the whole vehicle.
The calculation of temperature rise and distribution of air-cooled heat dissipation batteries lacks numerical introduction of heat transfer coefficients and convection heat transfer coefficients in batteries, cold plates and cold plates.
Author Response
The reviewer’s comments and suggestions are highly appreciated, and suggested modifications and improvements have been made in the revised paper. Point-to-point responses and discussions to the reviewer’s questions/suggestions are presented as follows:
1. Thank you for your suggestions. Simulating the battery under the actual working condition of the battery of the whole vehicle is one of the future works.
2. Thanks for your insightful comments. The thermal condition of the cell surfaces under the air-cooled simulation is “convection” with heat transfer coefficient 5 W/m2-K. A sentence was added in lines 319 to 320 as:
“The Thermal Condition for the battery cell walls is set to Convection with heat transfer coefficient 5 W/m2-K.”
Reviewer 3 Report
In this paper, the transient temperature distributions across a battery module consisting of four series-connected lithium-ion polymer battery cells are measured under various charging and discharging currents. A battery thermal model, correlated with the experimental data, in the module-level is built in the ANSYS/Fluent platform. And an air-cool thermal management system is integrated with the battery pack model to ensure the operating temperature and temperature gradient within the optimal range. This paper could provide thermal management design guideline for the lithium-ion polymer battery pack.
Amendments to the paper put forward:
(1) The temperature unit °K in the article is recommended to be modified to thermodynamic temperature K or unified to Celsius temperature °C;
(2) Some of the article font format is not correct, specifically in the last sentence of the second chapter;
(3) It is recommended to adjust the horizontal coordinate scale in Figure 2 according to the simulation time under different currents;
(4) The simulation results are somewhat unclear and the legend and unit distance are not suitable.
(5) In the article, copper is used as the material input of the tab. In fact, the positive and negative tab materials of the battery are different, which may has influence on the research results.
(6) References 19 do not clearly reflect the parameters related to the heat production of the battery. It is recommended to give the specific source of the battery parameters.
Author Response
The reviewer’s comments and suggestions are highly appreciated, and suggested modifications and improvements have been made in the revised paper. Point-to-point responses and discussions to the reviewer’s questions/suggestions are presented as follows:
1. Thank you for your suggestions. We have replaced all °K with K.
2. All incorrect font formats were modified.
3. The horizontal coordinates were adjusted in Fig. 2(a) and (b).
4. Thank you for your suggestions and we did try to improve the postprocessing results. The resolution of the ANSYS contour plots is not high enough and the color in the legend looks a little different from the color in the plot. This is an ANSYS/Fluent original issue and we could not improve it on our side.
5. The cathode and anode use different materials. However, the “tabs” in the ANSYS/Fluent battery model are current collectors, not the electrodes. Copper is the material for the current collectors for most lithium-ion battery cells. There is no modeling ANSYS/Fluent battery model does not model the cathode and anode.
6. We added one Reference 20 by an online ANSYS Fluent battery module manual that has detailed battery module introduction.
Round 2
Reviewer 1 Report
From line 151 to 191, you used many capital letters in sentence. Please correct.
-Also, you used many capital letters in sentence in other sections also.
-You mentioned that convective heat transfer coefficient is 5 W/m2-K with 2 m/s velocity.
-Is it reasonable guess for natural convection?
-This system looks like natural convection system, but you indicated flow inlet and outlet.
-In your manuscript, there are no space between numeric values and units. As you know some units need space or not, please check.
